# Exposure to Secondhand Heated-Tobacco-Product Aerosol May Cause Similar Incidence of Asthma Attack and Chest Pain to Secondhand Cigarette Exposure: The JASTIS 2019 Study

**DOI:** 10.3390/ijerph18041766

**Published:** 2021-02-11

**Authors:** Yuki Imura, Takahiro Tabuchi

**Affiliations:** 1School of Medicine, Osaka University, Osaka 565-0871, Japan; gtmwj864@yahoo.co.jp; 2Osaka International Cancer Institute, Osaka 541-8567, Japan

**Keywords:** heated tobacco product, secondhand smoke, asthma attack, chest pain

## Abstract

Although secondhand cigarette smoke is known to cause various health consequences, even the short-term effects of exposure to secondhand heated-tobacco-product (HTP) aerosol are unknown. The purpose of this study was to examine short-term symptoms related to secondhand HTP aerosol exposure. An internet-based self-reported questionnaire survey was conducted in 2019 as a part of the Japan Society and New Tobacco Internet Survey (JASTIS) study. In total, 8784 eligible respondents aged 15–73 years were analyzed. We examined the frequency (%) of secondhand combustible cigarette smoke and HTP aerosol exposure, and the exposure-related subjective symptoms (sore throat, cough, asthma attack, chest pain, eye pain, nausea, headache, and other symptoms). Overall, 56.8% of those exposed to secondhand cigarette smoke had any subjective symptoms, compared to 39.5% of those exposed to HTP aerosol. Asthma attack and chest pain were reported more frequently when associated with secondhand HTP exposure (10.9 and 11.8%, respectively) than with secondhand cigarette smoke exposure (8.4 and 9.9%, respectively). Sore throat, cough, eye pain, nausea, and headache were also more frequently reported when associated with secondhand cigarette smoke than with secondhand HTP exposure. This is the first study to examine severe subjective symptoms such as asthma attacks and chest pains, and to suggest that respiratory and cardiovascular abnormalities could be related to secondhand heated-tobacco-product aerosol exposure. Further careful investigations are necessary.

## 1. Introduction

Secondhand tobacco smoke can cause a variety of health consequences, such as respiratory symptoms, impaired lung function, and coronary heart disease [1,2]. Therefore, combustible cigarette smoking in public spaces has been prohibited [3]. In response to this social situation, tobacco companies have launched heated tobacco products (HTP) such as IQOS, glo, and Ploom TECH. HTPs are electronic devices that heat leaf tobacco. Users inhale aerosol containing tobacco extract instead of smoke; the aerosol is then exhaled into the surrounding air. Chemical analyses have shown that secondhand HTP aerosol may have fewer particular detriments than secondhand combustible cigarette smoke [4]. However, the degree of harmfulness has been unclear [3]. In Japan 2019, the percentage of people who smoked habitually was 16.7% [5], and that of HTP users was 11.3% [6]. Secondhand exposure will have affected more people.

Evaluation of the long-term effects of secondhand exposure on chronic disease is difficult because HTPs are recent products. Even short-term effects have seldom been reported [7]. Our previous survey only examined non-severe subjective symptoms such as sore throat, eye pain, and nausea. Therefore, our objective in this study is to examine short-term but relatively severe subjective symptoms, including chest pain and asthma attacks, in relation to secondhand HTP and combustible cigarette exposure. This is the first study to examine such symptoms.

## 2. Materials and Methods

### 2.1. Internet Survey

The Japan “Society and New Tobacco” Internet Survey (JASTIS) is a longitudinal internet-based cohort study which was designed to investigate perception, attitude, and use of HTPs, electronic cigarettes (e-cigarettes), and conventional tobacco products in Japan. The surveys were conducted between 2 and 28 February 2019. Respondents were selected from a large survey panel managed by a major nationwide internet research agency, Rakuten Insight [8], and were invited to participate in the survey. The survey was closed when the target number of respondents who had answered the questionnaire was reached. In the present study, we used cross-sectional data from the survey conducted among 11,000 people. Detailed information on the study has been provided in previous publications [7,9,10]. Panelists who consented to participate in the survey accessed the designated website and responded to the questionnaire. The survey data including questionnaires and details methods information can be accessed via the corresponding author on reasonable request.

### 2.2. Symptoms from Exposure to Secondhand Combustible Cigarette Smoke and HTP Aerosol

Participants were asked for self-reported experience of inhaling smoke produced by other people, using the following question: 

‘Have you inhaled the smoke of combustible cigarettes that other people were smoking within this one year?’ Response options were ‘never, 1–4 times, or 5 or more times.’

Later, those who had inhaled smoke produced by other people were asked for self-reported symptoms due to secondhand exposure, using the following questions:

‘Within this one year, have you experienced sore throat/cough/asthma attack/chest pain/eye pain/nausea/headache/other symptoms, after inhaling the smoke of combustible cigarette that other people were producing?’ Response options were ‘no, 1–4 times, or 5 or more times’.

Participants were also asked about secondhand HTP aerosol exposure using the questions that were applied to combustible cigarettes. Those who answered “1–4 times” or “5 or more times” were combined as the experienced group. Symptoms of sore throat, cough, asthma attack, and chest pain were combined and categorized as any respiratory symptom; although chest pain can be derived from lung and heart, it was assumed to be a respiratory symptom in this study. 

### 2.3. Characteristic Variables

Data for age group (15–19 years, 20–29, 30–39, 40–49, 50–59, 60–73), gender (man or woman), education (junior high school or high school, college or university and more), marital status (married, unmarried, and divorced or widowed), combustible-cigarette smoking status (non-smoker or current smoker), HTP using status (non-user or current user), and self-rated health (good or poor) were used as covariates. Married was defined as those who were married at the time of the survey; unmarried as those who had never married, and divorced or widowed. Current smoker or current HTP users were defined as those who had smoked or used HTPs in the previous 30 days. Self-rated health was dichotomized: good (excellent, very good, or good) or poor (fair or poor).

### 2.4. Statistical Analysis

Proportions of exposure to secondhand combustible cigarette and HTP and exposure-related subjective symptoms were calculated according to characteristic variables, comparing the proportions by fisher exact tests. Multivariable analyses were used to document the adjusted relationship between the above-mentioned variables and experienced symptoms. All analyses were performed using STATA/MP 15.0 for Windows (Stata Corp LLC, College Station, TX, USA).

## 3. Results

### 3.1. Percentages of Secondhand Exposure

A total of 472 respondents who had discrepancies in their responses were excluded, leaving 8784 subjects aged 15–73 years. Table 1 shows the basic characteristics of the study subjects and the percentages of exposure to secondhand combustible cigarette smoke and secondhand HTP aerosol. For example, 5142 subjects (58.5%) had been exposed to secondhand combustible cigarette smoke within the past year. Current smokers of combustible cigarettes or HTPs were more likely to be exposed to secondhand smoke or aerosols. Highly-educated (university or more) subjects were more likely to be exposed to secondhand smoke or aerosols than other groups. Subjects aged between 20–49 were more likely to be exposed to HTP aerosol than those aged over 50. 

### 3.2. Symptoms Associated with Secondhand Exposure

Table 2 and Table 3 and Figure 1 show the percentage (%) of subjects with self-reported symptoms associated with secondhand exposure. For example, 29.2% of those exposed to secondhand combustible cigarette smoke had experienced sore throat after inhaling the smoke and 23.0% of those exposed to secondhand HTPs aerosol had experienced sore throat after inhaling the aerosol. In total, 56.8% of those exposed to secondhand combustible cigarette smoke had any symptoms (sore throat, cough, asthma attack, chest pain, eye pain, nausea, headache, or other symptoms), compared with 39.5% of those exposed to HTP aerosol. The most common complaint was nausea (44.4% related to cigarettes and 31.9% by HTPs). Non-smoker and non-HTP-user participants were separately analyzed (results of the stratified analysis are shown in Figure 1B). Overall, 3995 non-smoker and non-HTP-users exposed to second-hand cigarette smoke, and 2102 persons exposed to second-hand HTP aerosol were analyzed and the results showed almost the same trends as the total sample. As a general trend, young people were likely to be more vulnerable than the elderly; those with poor self-related health were also likely to be more vulnerable than those with good health. Current smokers of cigarettes or HTPs were less likely to experience symptoms than non-smokers, except for asthma attack and chest pain which were reported more among current smokers.

## 4. Discussion

This is the first study to examine severe subjective symptoms such as asthma attacks and chest pains, which suggest respiratory and cardiovascular abnormalities related to secondhand HTP aerosol exposure. Surprisingly, compared to secondhand cigarette smoke, secondhand HTP aerosol exposure was more likely to be associated with asthma attacks and chest pains. On the other hand, in terms of relatively non-severe symptoms, sore throat and cough were less frequently related to secondhand HTP aerosol exposure than secondhand cigarette smoke, suggesting a small difference of approximately 20–30% between secondhand HTP and cigarettes exposure, which is consistent with our previous study [7].

Recent experimental studies indicate that HTP aerosol may be harmful, suggesting a possible mechanism. Of course, cigarette smoke can cause asthma and cardiovascular abnormalities including angina [2]. Exposure to cigarette smoke enhances sensitization to allergens and can bypass or override the normal tolerogenic response to inhaled antigen in mice [2]. Nicotine is vasoconstrictive [11], which may cause chest pain. Chemical analysis revealed that the amounts of harmful compounds, such as nicotine, in HTP aerosol were smaller than those in cigarette smoke [4,12,13]. However, it has not been confirmed that they were smaller than the threshold for attacks. It is known that even the smallest amount of tobacco smoke may cause toxic effects [2]. Moreover, several harmful compounds such as 2-furanmethanol were detected at higher levels in HTP aerosol than in smoke [13]. 2-furanmethanol is known as an irritant, thus, it might be related to asthma attacks and chest pain. An in vitro bioassay using respiratory cell lines suggested that higher concentrations of IQOS aerosol showed cytotoxicity equivalent to combustible cigarette smoke [14]. 

While HTPs operate differently than electronic cigarettes (e-cigarettes), reported health effects may be similar [15]. The aerosol from e-cigarettes has caused sore throat, cough, patho-physiological cardiovascular effects or airway obstructions [16,17]. An epidemiological study showed that adolescent e-cigarette users had increased rates of chronic respiratory symptoms [18]. Another study suggested increased odds of asthma among never combustible e-cigarette users [19]. Although it is unlikely to be associated with a single e-cigarette brand or compound, e-cigarette- or vaping-associated lung injury (EVALI) has been reported, at least suggesting that e-cigarettes can cause severe health outcomes including pneumonia and death [20]. 

This survey might reveal information about changes in the second-hand-exposure status of combustible cigarettes in recent years. For example, highly educated people reported where they encounter second-hand smoke, which was not seen in previous reports [21]. This may be because there are many large companies (employing educated people) in Japan that provide a separate area for smoking in the workplace rather than operating a complete smoking ban in large companies [22]. 

There are several limitations to the study. First, although the research agency seeks to ensure representativeness, the distribution is always imperfect; for example, there are no data of those who do not use internet. Additionally, self-reported information was used, and we excluded respondents with discrepancies or inconsistencies in the answers. Second, HTPs look like electronic cigarettes. Electronic cigarettes use liquid instead of tobacco leaf, and nicotine-containing liquid is prohibited in Japan. Some people might categorize HTPs, such as IQOS, as an electronic cigarette such as Juul, and vice versa. Therefore, some might have interchangeably selected both electronic cigarettes and HTPs, and this may have led to misclassification. Third, in some cases, the symptoms might be severe, others might recover rapidly. These situations may differ due to underlying comorbidities, which were not assessed in the survey. Such differences were not considered in this study, but will be included in future research. Similarly, the level of second-hand exposure could not be measured, thus a quantitative analysis could not be performed in this study. Further studies should better quantify exposure time and severity to evaluate a possible dose-response relationship. Fourth, because self-reported information is based on memory and subjective impression, there might issues of uncertainty. For example, the elderly are more likely to forget their experience, thus, the general trend that young people were likely to be more vulnerable than the elderly could be due, in part, to better recollection of their experience. Another example is that smokers or users of HTPs might not perceive symptoms as strongly because they do not perceive the harm from the products to be as severe since they use them. A previous study reported that adolescents who used electronic cigarettes had more favorable impression of them than those who did not use e-cigarettes [23]. Such problems of recall or user bias are possible. However, the results of non-smoker and no HTP user in this study (Figure 1B) were almost same to those of whole population (Figure 1A), suggesting that the user bias might not strongly influence the results. Fifth, several important issues could not be investigated in this survey. For example, the experience of acute symptoms related to second-hand exposure by others has been investigated, but the extent to which similar symptoms occurred with no second-hand exposure has not. Therefore, the degree to which second-hand exposure increases acute symptoms is unknown. This study examined experiences within one year, suggesting a risk of coincidental symptoms. Therefore, we will try next study using a definite period of time (for example one week or one month). Finally, this research used cross-sectional data, making it difficult to consider causal inference. The causal relationship between respiratory diseases and HTP should be clarified by longitudinal studies in the future.

## 5. Conclusions

HTPs are sometimes considered to pose lower health risks than cigarettes. However, asthma attack and chest pain were related to secondhand HTP exposure more frequently than secondhand cigarette smoke exposure. This is the first study to examine severe symptoms such as asthma attacks and chest pains, and to suggest that respiratory and cardiovascular abnormalities could be related to secondhand HTP aerosol exposure. Further study should better define and quantify this potential relationship.

## Figures and Tables

**Figure 1 ijerph-18-01766-f001:**
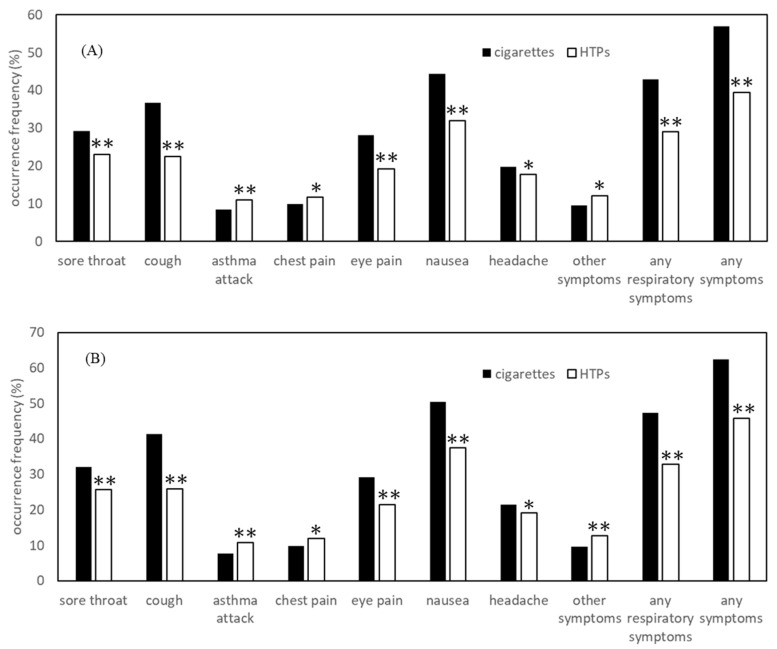
Occurrence ratio of symptoms from secondhand exposure among the whole sample (**A**), and among non-smoker and non-HTP-user participants (**B**). Occurrence ratio of sore throat, eye pain, nausea, cough was higher from secondhand cigarette smoke exposure than from secondhand HTP aerosol exposure. That of asthma attack and chest pain was higher from exposure to HTP aerosol than to cigarette smoke. * *p* < 0.05, ** *p* < 0.001.

**Table 1 ijerph-18-01766-t001:** Secondhand exposure according to characteristics of study subjects, the JASTIS 2019, *n* = 8784.

	Total		Secondhand Cigarette Smoke Exposure		Secondhand HTP Aerosol Exposure	
	*n*	%	*n*	%	*n*	%
Total	8784	100.0	5142	58.5	2923	33.3
Sex						
Man	4286	48.8	2583	60.3*	1606	37.5 **
Woman	4498	51.2	2559	56.9	1317	29.3
Age group, years						
15–19	853	9.7	483	56.6 **	278	32.6 **
20–29	1539	17.5	883	57.4	606	39.4
30–39	1373	15.6	807	58.8	527	38.4
40–49	1650	18.8	1002	60.7	603	36.5
50–59	1507	17.2	952	63.2	501	33.2
60–	1862	21.2	1015	54.5	408	21.9
Current cigarette smoker						
No	7482	85.2	4129	55.2 **	2267	30.3 **
Yes	1302	14.8	1013	77.8	656	50.4
Current HTP user						
No	7993	91.0	4554	57.0 **	2412	30.2 **
Yes	791	9.0	588	74.3	511	64.6
Education						
High school/less	2874	32.7	1616	56.2 **	868	30.2 **
2-year-college	1752	19.9	974	55.6	520	29.7
University/more	4158	47.3	2552	61.4	1535	36.9
Marital status						
Married	4492	51.1	2674	59.5	1461	32.5 *
Unmarried	3740	42.6	2158	57.7	1302	34.8
Divorced/widowed	552	6.3	310	56.2	160	29.0
Self-related health						
Good	7778	88.5	4568	58.7	2617	33.6 *
Poor	1006	11.5	574	57.1	306	30.4

Abbreviations: JASTIS, the Japan “Society and New Tobacco” Internet Survey; HTP, heated tobacco product. Note: *p* values for difference between statuses were obtained using chi-square or Fisher’s exact tests. * *p* < 0.05, ** *p* < 0.001 The * mark was only placed by the first covariate factor of the characteristic.

**Table 2 ijerph-18-01766-t002:** Respiratory symptoms related to cigarette smoke and HTP aerosol produced by others.

	Sore Throat		Cough		Asthma Attack		Chest Pain		Respiratory Symptoms		Any Symptom	
	Cig	HTP	Cig	HTP	Cig	HTP	Cig	HTP	Cig	HTP	Cig	HTP
Total	29.2	23.0	36.8	22.5	8.4	10.9	9.9	11.8	42.8	29.0	56.8	39.5
Sex												
Man	26.9 **	22.4	30.8 **	20.5 *	9.9 **	12.6 *	11.0 *	13.1 *	36.9 **	26.8*	49.6 **	35.6 **
Woman	31.5	23.6	42.8	25.0	6.8	8.9	8.7	10.3	48.8	31.6	64.2	44.2
Age group, years												
15–19	32.5 **	30.2 **	58.0 **	36.7 **	10.6 **	16.5 **	15.5 **	16.9 **	63.1 **	44.6 **	78.9 **	61.5 **
20–29	34.1	27.6	45.3	28.7	12.1	15.0	14.0	15.7	51.9	37.1	66.9	50.0
30–39	30.0	24.1	36.4	24.1	9.0	10.6	10.5	13.3	43.7	29.6	57.5	40.8
40–49	31.2	21.6	33.5	17.7	8.8	10.0	10.8	10.9	40.4	25.0	55.3	33.3
50–59	25.8	19.6	29.9	17.0	7.0	8.4	7.0	8.2	35.4	21.8	48.0	30.5
60–	23.7	15.9	29.2	15.7	4.3	6.1	4.7	6.4	34.0	20.1	46.9	27.2
Current cigarette smoker												
No	32.2 **	24.9 **	41.4 **	24.9 **	8.1	10.9	10.1	11.8	47.5 **	31.6 **	62.6 **	43.8 **
Yes	16.8	16.2	18.1	14.5	9.6	11.3	9.0	11.7	23.9	19.8	33.4	24.5
Current HTP user												
No	29.5	23.6	38.0 **	23.7 *	7.3 **	10.2 *	9.1 **	11.3	43.9 **	30.0 *	58.4 **	42.3 **
Yes	26.4	20.0	27.0	17.0	16.8	14.5	15.6	14.1	35.0	24.1	45.1	26.2
Education												
High school/less	29.8	24.4	39.7 *	25.1	9.4 *	12.9 *	10.7	12.9	44.9	31.6	58.2	42.5
2-year-college	27.6	22.9	35.2	21.5	6.4	8.5	8.4	11.9	40.2	27.3	54.1	36.3
University/more	29.3	22.1	35.5	21.4	8.5	10.7	9.9	11.1	42.5	28.1	57.0	38.8
Marital status												
Married	28.9	21.4 *	32.9 **	19.9 **	7.8 **	10.1 *	8.7 **	10.7 *	39.4 **	26.1 **	53.3 **	34.8 **
Unmarried	30.3	25.4	42.3	26.5	9.7	12.4	12.0	13.6	48.0	33.2	62.5	46.0
Divorced/widowed	23.9	17.5	31.3	14.4	4.2	6.9	5.2	7.5	36.5	20.6	48.7	28.8
Self-related health												
Good	27.9 **	22.0 *	35.6 **	21.7 **	7.8 **	10.5 *	9.4 *	11.3 *	41.7 **	28.2 **	56.3 *	38.7 **
Poor	39.0	31.0	45.8	29.7	13.1	15.0	13.6	16.0	51.6	35.6	61.3	46.1

Abbreviations: Cig; secondhand cigarette smoke exposure, HTP; secondhand heated-tobacco-product aerosol exposure. Note: *p* values for difference between statuses were obtained using chi-square or Fisher’s exact tests. * *p* < 0.05, ** *p* < 0.001 The * mark was only placed by the first covariate factor of the characteristic.

**Table 3 ijerph-18-01766-t003:** Non-respiratory symptoms related to cigarette smoke and HTP aerosol produced by others.

	Eye Pain		Nausea		Headache		Other Symptoms	
	Cig	HTP	Cig	HTP	Cig	HTP	Cig	HTP
Total	28.2	19.3	44.4	31.9	19.8	17.7	9.5	12.0
Sex								
Man	28.0	20.1	37.8 **	29.1 **	18.0 *	17.7	11.0 **	13.4 *
Woman	28.4	18.2	51.1	35.4	21.6	17.7	8.1	10.3
Age group, years								
15–19	32.9 **	23.4 **	65.8 **	50.4 **	32.1 **	26.6 **	12.2 **	15.8 **
20–29	30.6	23.8	52.1	39.4	25.0	23.6	12.5	16.0
30–39	30.7	21.6	45.0	34.3	22.3	19.2	10.0	12.0
40–49	32.9	18.6	42.7	27.9	20.9	16.6	10.8	11.3
50–59	24.6	15.0	38.3	24.8	16.6	12.8	8.1	10.2
60–	20.5	13.0	34.5	19.9	9.4	8.6	5.5	6.9
Current cigarette smoker								
No	29.7 **	20.7 **	50.4 **	35.8 **	21.8 **	18.7 *	10.0 *	12.5
Yes	22.1	14.3	20.0	18.6	11.7	14.3	7.8	10.2
Current HTP user								
No	27.9	19.7	46.1 **	34.5 **	19.7	17.7	8.9 **	11.7
Yes	30.6	17.2	31.6	20.0	20.6	17.8	14.3	13.5
Education								
High school/less	29.3	19.7	44.7	34.6	21.2	19.4	9.5	12.8
2-year-college	25.6	17.9	41.7	30.6	17.8	16.3	8.4	11.0
University/more	28.4	19.5	45.3	30.9	19.7	17.2	10.0	11.9
Marital status								
Married	27.0*	17.7 *	41.3 **	27.6 **	17.5 **	15.4 **	8.8 *	11.0
Unmarried	30.2	21.9	49.2	37.8	23.2	21.1	10.9	13.4
Divorced/widowed	24.8	12.5	38.1	23.8	15.8	10.6	6.8	10.0
Self-related health								
Good	27.4 **	18.5 *	43.7 *	31.1 *	19.1 *	16.7 **	9.2 *	11.6 *
Poor	34.5	25.5	50.3	38.6	25.3	26.5	12.4	15.7

Abbreviations: Cig; secondhand cigarette smoke exposure, HTP; secondhand heated-tobacco-product aerosol exposure. Note: *p* values for difference between statuses were obtained using chi-square or Fisher’s exact tests. * *p* < 0.05, ** *p* < 0.001 The * mark was only placed by the first covariate factor of the characteristic.

## Data Availability

The data used in this study are not available in a public repository because they contain personally identifiable or potentially sensitive patient information. Based on the regulations for ethical guidelines in Japan, the Research Ethics Committee of the Osaka International Cancer Institute has imposed restrictions on the dissemination of the data collected in this study. All data enquiries should be addressed to the person responsible for data management, Dr. Takahiro Tabuchi at the following e-mail address: tabuchitak@gmail.com.

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
