# Peer review of "Exposure to Secondhand Heated-Tobacco-Product Aerosol May Cause Similar Incidence of Asthma Attack and Chest Pain to Secondhand Cigarette Exposure: The JASTIS 2019 Study"

_ijerph, 2021, doi:10.3390/ijerph18041766_

Round 1
Reviewer 1 Report
The authors have significantly improved the original manuscript, addressing the issues detailed in the first review. I have only a few further suggestions, some of which are just stylistic editing.
Introduction
- Line 37: Would change “HTP aerosol has fewer particular detriments” to “HTP aerosol may have fewer…” or “HTP aerosol has been shown to have…” because this is based on one report, which I wouldn’t consider definitive.
- P 2 line 41: replace “such as” with “on”
Material/Methods
- Section 2.2: are the questions cited here excerpts from the survey? Or the whole survey? It may be helpful to include the survey as a supplemental file.
- Section 2.3: Were HTP also characterized as former user? Or former smoker (more likely since HTP are so new). I ask because you may want to address this in the limitations section (see my comment later regarding bias).
Results
- Section 3.2 lines 119-121: That current smokers of cigarettes and/or HTPs were less likely to experience symptoms than non-smokers is very interesting and may speak to the question of bias in the response. That is, perhaps smokers/users of HTP do not perceive symptoms as strongly because they do not perceive the harm from the products to be as severe since they use them. See comments re limitations and perception of harm, later. Note this reference – which surveyed adolescents and found that those who used e-cigarettes had more favorable impression (less harm) of them than those who didn’t use ecigs - Gorukanti A, Delucchi K, Ling P, Fisher-Travis R, Halpern-Felsher B. Adolescents' attitudes towards e-cigarette ingredients, safety, addictive properties, social norms, and regulation. Prev Med. 2017 Jan;94:65-71. doi: 10.1016/j.ypmed.2016.10.019. Epub 2016 Oct 20. PMID: 27773711; PMCID: PMC5373091.
Discussion
- P 8 line 158: I would leave out the statement that HTP are considered more harmful than electronic cigarettes. Not sure that is universally true. Perhaps just state, “While HTP operate differently than e-cigarettes, reported health effects may be similar.” Or something like that.
- P 8 line 167: replace “have more situations” with “reported”
- Line 170 – 171: delete the last phrase “ these companies employ…” and add in the phrase, in parentheses, in line, 169 – “this may be because there are many large companies (employing educated people)….”
- Line 173: “The distribution is always imperfect.” Can you clarify what this means? It may be skewed by zip code? How might it be imperfect?
- Line 180: I would specify, after the phrase, “due to underlying comorbidities” to add the phrase, “…comorbidities, which were not assessed in the survey.”
- Line 182 – 183: I would replace “further sophisticated study design…future” with something like, “further studies should better quantify exposure time to evaluate a possible dose-response relationship.
- Line 184: the concept about the elderly and memory is interesting. I would expand on the concept of recall and user bias – that those who use products may be less likely to experience symptoms or may perceive use to be less dangerous (thereby perceiving less harm) – see the reference I provided earlier. Also, I don’t think the survey asked about HTP users who were former smokers (or vice versa) but that might also influence attitudes about the products. It would be interesting to assess the respondents’ concept of harm (How harmful do you think each product is) and compare this with reported symptoms.
Conclusions
- I would delete the last line; it’s redundant. Consider adding a line like, “Further study should better define and quantify this potential relationship.
Author Response
Reviewer1
Comments to the Author:
The authors have significantly improved the original manuscript, addressing the issues detailed in the first review. I have only a few further suggestions, some of which are just stylistic editing.
#1
Line 37: Would change “HTP aerosol has fewer particular detriments” to “HTP aerosol may have fewer…” or “HTP aerosol has been shown to have…” because this is based on one report, which I wouldn’t consider definitive.
Our response: We have modified the sentence.
Change (Introduction):
Chemical analyses have shown that secondhand HTP aerosol may have fewer particular detriments than secondhand combustible cigarette smoke [4].
#2
P 2 line 41: replace “such as” with “on”
Our response: We have modified the sentence.
Change (Introduction):
Evaluation of the long-term effects of secondhand exposure on chronic disease is difficult because HTPs are recent products.
#3
Section 2.2: are the questions cited here excerpts from the survey? Or the whole survey? It may be helpful to include the survey as a supplemental file.
Our response: They are excerpts from the survey. The survey data including questionnaires and details methods information can be accessed via the corresponding author on reasonable request.
Change (Materials and Methods):
The survey data including questionnaires and details methods information can be accessed via the corresponding author on reasonable request.
#4
Section 2.3: Were HTP also characterized as former user? Or former smoker (more likely since HTP are so new). I ask because you may want to address this in the limitations section (see my comment later regarding bias).
Our response: We did not characterize former user of HTP or former smoker in this study. Please see our response to your comment #12.
#5
Section 3.2 lines 119-121: That current smokers of cigarettes and/or HTPs were less likely to experience symptoms than non-smokers is very interesting and may speak to the question of bias in the response. That is, perhaps smokers/users of HTP do not perceive symptoms as strongly because they do not perceive the harm from the products to be as severe since they use them. See comments re limitations and perception of harm, later. Note this reference – which surveyed adolescents and found that those who used e-cigarettes had more favorable impression (less harm) of them than those who didn’t use ecigs - Gorukanti A, Delucchi K, Ling P, Fisher-Travis R, Halpern-Felsher B. Adolescents' attitudes towards e-cigarette ingredients, safety, addictive properties, social norms, and regulation. Prev Med. 2017 Jan;94:65-71. doi: 10.1016/j.ypmed.2016.10.019. Epub 2016 Oct 20. PMID: 27773711; PMCID: PMC5373091.
Our response: We added some descriptions in the discussion section. Please see our response to your comment #12.
#6
P 8 line 158: I would leave out the statement that HTP are considered more harmful than electronic cigarettes. Not sure that is universally true. Perhaps just state, “While HTP operate differently than e-cigarettes, reported health effects may be similar.” Or something like that.
Our response: We have modified the sentence.
Change (Discussion):
While HTPs operate differently than e-cigarettes, reported health effects may be similar
#7
P 8 line 167: replace “have more situations” with “reported”
Our response: We have modified the phrase.
Change (Discussion):
For example, highly educated people reported where they encounter second-hand smoke, which was not seen in previous reports [21].
#8
Line 170 – 171: delete the last phrase “these companies employ…” and add in the phrase, in parentheses, in line, 169 – “this may be because there are many large companies (employing educated people)….”
Our response: We have modified the phrase.
Change (Discussion):
This may be because there are many large companies (employing educated people) in Japan that provide a separate area for smoking in the workplace rather than operating a complete smoking ban in large companies [22].
#9
Line 173: “The distribution is always imperfect.” Can you clarify what this means? It may be skewed by zip code? How might it be imperfect?
Our response: We have added more information about imperfectness.
Change (Discussion):
First, although the research agency seeks to ensure representativeness, the distribution is always imperfect; for example, there are no data of those who do not use internet.
#10
Line 180: I would specify, after the phrase, “due to underlying comorbidities” to add the phrase, “…comorbidities, which were not assessed in the survey.”
Our response: We have modified the sentence.
Change (Discussion):
These situations may differ due to underlying comorbidities, which were not assessed in the survey.
#11
Line 182 – 183: I would replace “further sophisticated study design…future” with something like, “further studies should better quantify exposure time to evaluate a possible dose-response relationship.
Our response: We have modified the sentence.
Change (Discussion):
Further studies should better quantify exposure time and severity to evaluate a possible dose-response relationship.
#12
Line 184: the concept about the elderly and memory is interesting. I would expand on the concept of recall and user bias – that those who use products may be less likely to experience symptoms or may perceive use to be less dangerous (thereby perceiving less harm) – see the reference I provided earlier. Also, I don’t think the survey asked about HTP users who were former smokers (or vice versa) but that might also influence attitudes about the products. It would be interesting to assess the respondents’ concept of harm (How harmful do you think each product is) and compare this with reported symptoms.
Our response: We really appreciate your comments. Also related to your comments #4 and #5, we added some descriptions in the discussion section. although user bias might be small considering Fig. 1(B). The results of non-smoker and no HTP user in this study were almost the same to those of whole population.
Change (Discussion):
Fourth, because self-reported information is based on memory and subjective impression, there might issues of uncertainty. For example, the elderly are more likely to forget their experience, thus, the general trend that young people were likely to be more vulnerable than the elderly could be due, in part, to better recollection of their experience. Another example is that smokers or users of HTPs might not perceive symptoms as strongly because they do not perceive the harm from the products to be as severe since they use them. A previous study reported that adolescents who used electronic cigarettes had more favorable impression of them than those who didn’t use e-cigarettes [23]. Such problems of recall or user bias are possible. However, the results of non-smoker and no HTP user in this study (Figure 1 (B)) were almost same to those of whole population (Figure 1 (A)), suggesting that the user bias might not strongly influence the results.
Change (References):
- Gorukanti, A.; Delucchi, K.; Ling, P.; Fisher-Travis, R.; Halpern-Felsher, B. Adolescents' attitudes towards e-cigarette ingredients, safety, addictive properties, social norms, and regulation. Prev Med 2017, 94, 65-71. doi: 10.1016/j.ypmed.2016.10.019.
#13
I would delete the last line; it’s redundant. Consider adding a line like, “Further study should better define and quantify this potential relationship.
Our response: We have deleted the line and added the sentence as follows.
Change (Discussion):
Further study should better define and quantify this potential relationship.

Reviewer 2 Report
The authors have addressed my second point, but have not addressed my first – that the questions asked both “prompt” participants to recall symptoms that they did not necessarily experience, and that they ask about “ever” exposure rather than exposure in a defined period of time (which obviously expands the amount of time for coincidental symptoms to occur). There is no way to resolve these concerns and thus the paper should be rejected.
Author Response
Reviewer 2
The authors have addressed my second point, but have not addressed my first – that the questions asked both “prompt” participants to recall symptoms that they did not necessarily experience, and that they ask about “ever” exposure rather than exposure in a defined period of time (which obviously expands the amount of time for coincidental symptoms to occur). There is no way to resolve these concerns and thus the paper should be rejected.
Our response:
We are very sorry for mistranslation of our questionnaire and make you confuse about that. We corrected the descriptions of the questionnaire as follows; an experience of symptoms within one year, not entire life.
BEFORE
‘Have you ever experienced sore throat/ cough/ asthma attack/ chest pain/ eye pain/ nausea/ headache/ other symptoms, after inhaling the smoke of combustible cigarette that other people were producing?’
AFTER CORRECTION
‘Within this one year, have you experienced…’
This was a severe mistranslation. We modified the sentence.
One year may be long to recall experienced symptoms and there was an increased risk to occur coincidental symptoms. We perform surveys every year, thus the study used a one-year period to estimate the cumulative incidence of the exposure. We added some descriptions in the limitation section as follows.
Change (Materials and Methods):
Participants were asked for self-reported experience of inhaling smoke produced by other people, using the following question:
‘Have you inhaled the smoke of combustible cigarettes that other people were smoking within this one year?’ Response options were ‘never, 1-4 times, or 5 or more times.’
Later, those who had inhaled smoke produced by other people were asked for self-reported symptoms due to secondhand exposure, using the following questions:
‘Within this one year, have you experienced sore throat/ cough/ asthma attack/ chest pain/ eye pain/ nausea/ headache/ other symptoms, after inhaling the smoke of combustible cigarette that other people were producing?’ Response options were ‘no, 1-4 times, or 5 or more times.’
Change (Discussion):
Fifth, several important issues could not be investigated in this survey. For example, the experience of acute symptoms related to second-hand exposure by others has been investigated, but the extent to which similar symptoms occurred with no second-hand exposure has not. Therefore, the degree to which second-hand exposure increases acute symptoms is unknown. This study examined experiences within one year, suggesting a risk of coincidental symptoms. Therefore, we will try next study using a definite period of time (for example one week or one month).

Reviewer 3 Report
You did a good job responding to prior comments.
Author Response
Reviewer 3
You did a good job responding to prior comments.
Our response: We really appreciate your suggestions. Thanks a lot.

Round 2
Reviewer 2 Report
I thank the authors for correcting this mistranslation, but this doesn't address the underlying point. Recall over a year is far too long a period to link to acute symptoms. Recall will not be accurate which will bias the results in unpredictable ways. This methodology is not suitable to answer the research question. I remain concerned that participants were prompted by the question design. I continue to recommend rejection.Author Response
We are very sorry that our study has a limitation for recall bias. We recognize that the period of one year is a risk of the recall bias, so we had already written this point in the limitation section as “This study examined experiences within one year, suggesting a risk of coincidental symptoms. Therefore, we will try next study using a definite period of time (for example one week or one month).”
While we do not think that this study alone can draw conclusions, we believe that this study may be valuable as one of the data for discussion.